# Nutrition Patterns and Their Gender Differences among Rheumatoid Arthritis Patients: A Descriptive Study

**DOI:** 10.3390/nu15010095

**Published:** 2022-12-24

**Authors:** Christina Heidt, Ulrike Kämmerer, Thorsten Marquardt, Monika Reuss-Borst

**Affiliations:** 1University of Muenster, D-48149 Muenster, Germany; 2Department of Obstetrics and Gynaecology, University Hospital of Wuerzburg, D-97080 Wuerzburg, Germany; 3Department of General Pediatrics, Metabolic Diseases, University of Muenster, Albert-Schweitzer-Campus, D-48149 Muenster, Germany; 4Center for Rehabilitation and Prevention Bad Bocklet, D-97708 Bad Bocklet, Germany; 5Department of Nephrology and Rheumatology, Georg-August University of Goettingen, D-37075 Goettingen, Germany

**Keywords:** rheumatoid arthritis, nutritional patterns, diet quality, dietary guidelines, nutrition knowledge, lifestyle advice, multidisciplinary team

## Abstract

Dietary factors probably play a role in the pathogenesis and clinical course of rheumatoid arthritis (RA). There is a paucity of specific dietary guidelines for RA patients and little information on their implementation in daily life. Therefore, this study aimed to determine the nutritional status and provision of nutritional education among outpatients with RA. Here, 61 patients were included with a sex ratio of 2.03 (f/m). Based on BMI, 22% of women were overweight and 32% obese, whereas 50% of men were overweight and 30% obese. Fasting blood and a 3-day estimated dietary record were collected. Additionally, patients were asked whether they had already received information about a specific diet as part of their disease treatment plan. Elevated total cholesterol levels were found in 76% of women and in 60% of men caused by increased non-HDL-C levels. The dietary intake assessment showed a lower self-reported intake of energy, polyunsaturated fat, carbohydrates, fiber, and several micronutrients than recommended. Regarding healthy eating, all patients reported familiarity with dietary recommendations, but found it difficult to implement the recommendations into their diets. These findings suggested that RA patients need more specific recommendations and education in clinical practice to improve the quality of their diet.

## 1. Introduction

Rheumatoid arthritis (RA) is a chronic autoimmune disease. Characteristics of this disease include chronic inflammation of the synovial tissues resulting in joint swelling, stiffness, pain, and destruction of cartilage [1]. The prevalence of diagnosed RA in adults in Germany is 1.26% [2]. The pathogenesis of RA is not yet fully understood, though it is known that both genetic and environmental factors are involved [3,4,5,6]. Evidence suggests that dietary factors might act as important epigenetic triggers leading to the development of RA [7]. In particular, a Western diet, characterized by increased consumption of saturated fats and oils rich in omega-6 fatty acids, decreased consumption of omega-3 fatty acids, and increased consumption of refined carbohydrates as well as sugar-sweetened drinks, raises the risk for RA [8,9]. This dietary pattern predisposes to the risk of RA both directly, by increasing inflammation, and indirectly, through increasing insulin resistance, obesity, diabetes, and cardiovascular diseases along with accelerated atherosclerosis, and dementia [10,11,12,13,14,15,16]. Emerging literature suggests that healthy dietary habits may have a protective role against the development and progression of RA and associated co-morbidities [17,18]. Presently, there are no specific dietary guidelines for patients with RA apart from omega-3 supplementation [19]. However, in addition to drug treatment of RA, the German patient association Rheuma-Liga recommends its members or patients consume a healthy diet based on the 10 guidelines of the German Nutrition Society (DGE) for a wholesome diet [20]. In addition to these guidelines, the Rheuma-Liga suggests a healthy diet including vegetables, fruits, whole grains, nuts, seeds, herbs, spices, omega-3-rich foods (e.g., fish and healthy oils), and a diet low in saturated fat, arachidonic acid from red meat and milk products, and refined carbohydrates, as shown in Figure 1 [21].

In a review on that topic, 227 articles on the effectiveness of diet in patients with RA were evaluated [22]. Generated from strength of evidence and conclusions, the authors developed a food pyramid allowing patients with RA to easily understand what best to eat and how to improve their eating habits [22,23].

Recently, short-chain fatty acids (SCFAs) have been shown to influence the development and progression of RA [24,25,26]. SCFAs (e.g., butyrate, acetate, and propionate) are the main metabolic products of anaerobic bacterial fiber fermentation in the intestine and have many effects on host physiology such as acting as an energy source for colonocytes, regulating the gut barrier, and influencing inflammatory response and immunity [27]. A high-fiber diet increases SCFA levels and decreases the inflammatory response in patients with RA [28]. Furthermore, an increased intake of dietary fiber significantly improved the physical function and quality of life in RA patients [24,25,28]. It appears likely that a healthy, high-fiber diet may be a game-changer for patients with a high risk of developing RA. Therefore, dietary education and monitoring of the diet and nutritional status by rheumatologists could help to control inflammation and to improve overall health and quality of life in their RA patients. However, to date, the dietary patterns of RA patients have not been evaluated and are largely unknown [23].

The aim of this study was therefore to collect baseline information on RA patients regarding nutritional status, food consumption, whether they had received nutritional education.

## 2. Materials and Methods

### 2.1. Study Design and Patients

The patients participating in this analysis were screened at enrollment in an ongoing single-centre randomized, double-blind, controlled clinical study that conformed to the Declaration of Helsinki (current version 2013) and Good Clinical Practice Guidelines. The study’ s protocol was approved by Bavarian Ethics Committee (approval number: 21020), and was registered at the German Registry of Clinical Trials (DRKS00025413). For the current study, we enrolled 61 patients fulfilling the RA classification criteria of the American College of Rheumatology/European League Against Rheumatism (ACR/EULAR) at the rheumatology outpatient practice Bad Bocklet, Germany [29]. The characteristics of the patients are given in Table 1. The study was conducted from August 2021 to October 2022. All patients provided their informed written consent.

### 2.2. Anthropometric Measurements

Anthropometric measurements were carried out in each of the patients, including body height and weight as well as waist and hip circumferences. Weight was measured to the nearest 0.1 kg using a digital scale while height was measured to the nearest 0.1 cm using a stadiometer. Based on the results of anthropometric measurements, Body Mass Index (BMI) and Waist-to-Hip Ratio (WHR) were calculated. BMI was calculated as weight divided by height squared, WHR—by dividing waist circumference by hip circumference. BMI results were interpreted using WHO classification. For waist circumference and WHR, cut-off points developed by WHO were used [30,31].

### 2.3. Disease Activity

Disease activity was evaluated by SDAI (Simple Disease Activity Index), measured by the arithmetic sum of tender and swollen 28-joint count, the patient’s and rheumatologist’s global assessment, and CRP in mg/dL [32].

### 2.4. Biochemical Analysis

A Cobas 4000 analyzer (Roche Diagnostics, Mannheim, Germany) was used to measure triglycerides (TG), total cholesterol (TC) and HDL-C. Blood samples for the analysis of biochemical parameters were collected after at least 8 h of fasting. Serum tubes (BD Vacutainer, 5.0 mL) were centrifuged at 700× *g* for 10 min at room temperature. TC, TG, HDL-C were measured directly in fresh serum samples. All analyses were performed within 90 min from blood collection. TG, TC and HDL-C were determined enzymatically via **P**henol and **A**mino**p**henazone (PAP)-methods. LDL-C values were calculated by using the Friedewald formula (LDL [mg/dL] = TC [mg/dL] – HDL [mg/dL] – Triglyceride [mg/dL]/5) [33]. Non-HDL-C was calculated as the difference between TC and HDL-C [34]. Table 2 shows anthropometric and biochemical values.

### 2.5. Dietary Intake Analysis

Patients were asked to complete a validated 3-day estimated dietary record (two consecutive days and a day-off), which listed 146 food items, subdivided into 16 food groups. For every food item both typical household measures (e.g., slice, cup, spoon) and the appropriate weights were provided in a record booklet. Based on these information, patients were supposed to estimate the amount of their food consumption [35]. In addition, patients were asked to provide information on nutritional habits by using a questionnaire. The nutrition questionnaire covered quantitative information about how often the patients consumed certain foods and beverages: bread, potatoes, pasta, rice, fruits, vegetables, legumes, nuts, cereals, meat, fish, eggs, dairy products, fat and oils, confectionary, pastries, and non-alcoholic beverages. A trained nutritionist evaluated the patients’ dietary intake and questionnaires. Food items were entered into a customized nutritionist software PRODI^®^ 6 expert (Nutri-Science GmbH, Freiburg, Germany) and mean intake of energy, macro- and micronutrients and food group intakes were calculated. German dietary guidelines 2015 of the German Nutrition Society (DGE) were used to compare the reported results [20]. Information on nutritional supplement use was collected during the screening visit, but data were not included in the food record analysis.

### 2.6. Nutritional Education

Aside from dietary intake, the patients were asked whether they had already received information about a healthy diet as part of their disease treatment plan, and, if so in what context. Possible answers were “yes” or “no”, based on medical advice, detailed advice from a nutritionist, advice from Rheuma-Liga by written information, e.g., brochures, leaflets. Specific information herein describes a healthy diet to include vegetables, fruits, whole grains, nuts, seeds, herbs, spices, omega-3-rich foods (e.g., fish and healthy oils), and a diet low in saturated fat, arachidonic acid from red meat and milk products, and refined carbohydrates, as shown in Figure 1.

### 2.7. Statistical Analysis

GraphPad Prism version 9.3.1 (GraphPad, La Jolla, CA, USA) was used to analyze data. Data are presented as means, standard deviations (SD), medians and interquartile range (IQR). Normality of the analyzed variables were tested with the Kolmogorov–Smirnov test at the level of *p* < 0.05. Since the distributions varied from normality, to compare the values of analyzed parameters within the given sex between two groups, a nonparametric Mann–Whitney-U test was used. The level of statistical significance was set at *p* < 0.05. Spearman’ s correlation test was used for correlations. *p* < 0.05 was considered to indicated a statistically significant difference (see Appendix A).

## 3. Results

### 3.1. Patient Characteristics

A total of 61 patients (67% female, 33% male) were enrolled in the study. Demographic and clinical characteristics by sex are presented in Table 1. The nutritional status was assessed on the basis of anthropometric measurements: body height and weight (Body Mass Index, BMI) as well as waist and hip circumferences (Waist-to-Hip-Ratio) and biochemical values. Waist circumference was also analyzed with regard to increased risk of metabolic complications.

### 3.2. Dietary Intake

Energy and nutrition intake calculated from patient’s 3-day-food records are provided in Table 3. Two females (4.9%) were vegetarians. On average, energy intake was 1492 ± 417 kcal for females, with 45% of kcal from carbohydrate intake, 15% of kcal from protein intake, and 40% from fat intake. The energy intake for males was 1685 ± 588 kcal, with 48% of kcal from carbohydrate intake, 16% of kcal from protein intake, and 36% from fat intake. The mean intake of polyunsaturated fatty acids was found to be below the recommendations for females and males (Table 3). The average consumption of linoleic acid and linolenic acid of examined female patients was in accordance with the recommendations and amounted to 7.3 ± 0.6 g (152% of the norm) and 1.2 ± 1.3 g per day (125% of the norm). The average content of linoleic acid in the men’s diet was similar to that of the women’s, however, a lower intake of linolenic acid was noted in men (74 of norm). Moreover, reported cholesterol, salt and alcohol intake were in line with recommendations for most patients. Analysis of the food records showed low consumption of dietary fiber for both males and females. 98% of females and males consumed less than 30 g fiber /day (Table 4). The mean daily consumption of caffeine (638.7 ± 493.6; corresponding to 6–7 cups of coffee and/or black/green tea), was significantly higher in men than in women [36].

### 3.3. Micronutrient Intake

In patients’ food records, the contents of almost all analyzed micronutrients were low, except sodium and copper, in men and women, as shown in Table 5. Statistical analysis showed significantly lower sodium content in diets of women than of men. Among males, mean vitamins A, and K intakes were below the recommendations. For vitamin D, 98% of females and males did not reach the recommendations. Females and males also consumed vitamin E, B6, folate, calcium, magnesium, zinc and iron below diet recommendations (Table 5).

### 3.4. Nutrition Patterns

The detailed consumption frequencies of food groups by sex are presented in Table 6. On average men consumed significantly more bread than women. Furthermore, men consumed more potatoes, milk, dairy products, meat and fish, whereas the consumption of vegetables, fruits, nuts, cheese, and eggs was higher in women. However, for both females and males, the mean reported intake of vegetables and fruits was below the recommendations. Among women, intake of butter, margarine, and oils was significantly higher.

### 3.5. Nutritional Education

All patients received advice on healthy eating from the treating rheumatologist. A total of 80% of all patients stated that they had received written nutrition information, e.g., brochures, leaflets. None of the included patients reported that they had received detailed and individualized advice from a nutritionist.

### 3.6. Correlations between Anthropometric, Biochemical Parameters and Nutrition Intake

Correlations by sex are presented in Appendix A.

## 4. Discussion

To our knowledge, this is the first study with the primary goal of investigating the nutrient intake and nutrition patterns of German patients with RA. One major finding of our study is that a large percentage (72.4%) of our study population reported inadequate intake of several nutrients (macro- and micronutrients) that were below the dietary guideline values of the German Society of Nutrition (DGE). In addition, we found that patients with RA, in general, consume less plant-based food such as vegetables, fruits, nuts, and seeds, and higher amounts of foods of animal origin than recommended. The study also found that, while all included patients received advice on healthy eating from the treating rheumatologist, the patients did not receive individualized detailed advice on how to meet daily recommended intakes of fruits, vegetables, healthy fats, whole grains, low-fat dairy, and omega-3-rich fish.

In our study, we found that the intake of macronutrients such as energy, polyunsaturated fat, and carbohydrates was generally lower than recommended. Our results were similar to the recently published Swedish study ADIRA that assessed the energy and nutrient intake in females and males with RA, using a 3-day food record [37]. Similar results were reported by other authors as well [38,39]. Other studies have mostly reported a different macronutrient distribution compared to our study, including higher energy, carbohydrate, fat, sugar, and arachidonic acid intakes [40,41,42,43,44]. Under-reporting of energy intake and reporting of biased dietary information has been described in individuals with RA, which may have influenced average energy and macronutrient intake [38,45]. Additionally, several food consumption studies found a similar level of underreporting [46,47,48]. The degree of under-reporting is greater in obese subjects [49,50,51,52]. Results from the longitudinal study on nutrition and health status in an aging population in Giessen, Germany (GISELA), using the same 3-day food record, which was developed and validated for that study, confirmed that volunteers with a heavy body weight, and large BMI and fat mass underreport their energy intake [52].

As for micronutrients, we found that the reported intakes were especially low for vitamin D, E, B6, folic acid, calcium, magnesium, iron, and zinc. This could be due to generally lower consumption of vegetables, legumes, fruits, bread, cereals, eggs, cheese, butter, and oils by women and men. Additionally, our study found that the reported intake of vitamin A and K was low among the male population due to lower consumption of vegetables, legumes, fruits, eggs, cheese, butter, and oils than in females. For this study, we did not include supplements in the food records. However, the majority of enrolled patients reported taking prescribed folic acid and/or vitamin D to overcome medication side effects and also consumed other supplements. Therefore, some patients may have achieved the daily recommended nutrient intakes by taking supplements. Similar results have been reported in a few studies [37,53,54]. However, some studies display a different micronutrient distribution compared to our study [39,45,55]. This could be due different dietary recommendations or dietary strategies between the countries. Nevertheless, it can be assumed that the 3-day estimated dietary record could neither underestimate nor overestimate dietary intake of micronutrients [35].

High-fiber diets are beneficial and reduce the risk of obesity, diabetes, hypertension, and cardiovascular disease (CVD), which are common comorbidities of RA [15,16,56]. In patients with RA, approx. 50% of deaths are attributable to CVD-related causes [57]. Notably, data reveal an increased risk in RA patients of cardiovascular mortality, which correlates negatively with dietary fiber intake [58]. A recent study of 183 patients with RA from Iran found that a low-fiber dietary pattern was linked to accelerated RA disease development. In our study, the mean intakes for fiber were lower than current recommendations due to lower consumption of vegetables, fruits, and cereals. Reported fiber intakes were not higher in other studies for people with RA or in the general German population [38,59,60]. Dietary fiber supplementation has a potential to close this fiber gap [56]. Interestingly, none of the patients used a dietary fiber supplement or asked for information on high-fiber food sources. This could be due to a lack of precise disease-related recommendations for RA as well insufficient information on the preventive potential of fiber.

Around 60% of patients with RA are either overweight or obese (defined as having a BMI ≥ 25kg/m^2^ and ≥ 30kg/m^2^, respectively), which is comparable to the general population [61,62]. Similar results were obtained by our group. We found that 22% of women were overweight and 32% obese, whereas 50% of men were overweight and 30% obese. Overweight and obesity are strong risk factors for the development of CVD in RA [15]. No evidence exists that a lower BMI or modifying body composition in patients with RA will reduce CVD morbidity and mortality (obesity paradox) [63]. One hypothesis is that the BMI is a poor proxy for the nutritional and metabolic state of patients with RA, as reduced lean mass in these patients might mask increased body fat percentages [64]. However, it seems logical that, as for the general population, an increase in lean mass and a reduction in fat percentage would incur beneficial effects on CVD risk in patients with RA [63]. Numerous studies have provided consistent evidence for a causal relationship between blood cholesterol concentrations and cardiovascular disease [65]. With regard to TC and non-HDL-C, both established indicators of CVD risk, we found increased TC levels (the mean intake of dietary cholesterol was in accordance with the recommendations for most patients) as well as increased non-HDL-C levels in women and men. As with the general population, patients with RA also have an increased risk of CVD when lipid levels are either too low or too high. Lipid abnormalities are quite commonly present in patients with RA [66]. Roughly one third of RA patients have hypercholesterolemia [67,68]. Lifestyle interventions should be advised to achieve guideline-recommended treatment targets. The European Guideline for prevention of cardiovascular disease recommends a daily intake of 200 g fruits and 200 g vegetables, divided into 2–3 servings throughout the day [69]. For metabolic syndrome there are several recommendations such as Mediterranean Diet, Dietary Approaches to Stop Hypertension (DASH) diet, plant-based diets, or fasting [70]. Taken altogether, the average intake of vegetables and fruits was lower than recommended and, moreover, 3% of patients were on a plant-based diet, which is defined by a preferred intake of vegetables, fruits, legumes, seeds, nuts, whole grain products, and vegetable oils. This could be due to a lack of clear recommendations in practice guidelines and in nutrition advice. Individualized dietary advice by a nutritionist or dietician could help patients to easily implement the concrete recommendations.

In the present study, we observed differences in food consumption between the sexes. Men showed a significantly higher consumption of bread, caffeine, and sodium than women. On average, men consumed more meat, fish, milk, and milk products, potatoes, and three times more alcoholic beverages than women. In women, consumption of fruits, vegetables, cheese, eggs, fats, and oils was higher. The German National Nutrition Survey II (NVS II) showed similar sex-related differences in food consumption, especially for bread, potatoes, and water in the German population residing in Bavaria. Furthermore, we found differences in the consumption of milk and milk products, cheese, and fruits in women and men, and less consumption of milk and milk products, cheese, and fruit but more vegetable and fish consumption by women and men in comparison to the NVS II. In men, consumption of fish was 2.2 and, in women, 1.1 times higher than in NVS II participants [71]. Consumption of vegetables among both populations was 1.2 times higher. Interestingly, we observed that, in men, the consumption of meat and meat products was 1.4 times lower than in NVS II participants, suggesting that patients with RA make healthier food choices than NVS II participants likely due to nutritional guidance provided by rheumatologists and adherence to general dietary recommendations [72,73]. More and better dietary advice may result in higher consumption of nutritionally important foods such as fish and vegetables. Garner et al. demonstrated similar data on individualized counselling as a dietary intervention and subsequent improved dietary intakes [74]. Taken together, it has been recognized that patients appreciate education from nutritionists above and beyond education by the physicians [75]. Given the solid evidence for negative consequences of an unhealthy diet on disease activity and the preventive potential of a healthy diet, it appears reasonable to integrate nutritionists or dieticians in the multidisciplinary team as already established in Sweden, Denmark, Norway, Netherlands, and partially in the UK [76,77]. The current study has some strengths but also limitations. Strengths of our study include the use of a 3-day estimated dietary record which was developed and validated for the GISELA study [31]. Dietary intake was assessed and reviewed by a trained nutritionist for completeness. Blood samples were taken for RA disease activity index (SDAI) and further biochemical RA-relevant parameters. We included women and men with a sex ratio of 2.03 (f/m), usually women are threefold more likely to develop RA than men [78]. Limitations include the use of self-reported and possibly under-reported and or over-reported dietary data. We could not rate the consumption of supplements in this study, since none of the patients re-corded any intake of supplements in the 3-day-food records. In addition, we asked yes-no questions about the intake of vitamins and mineral supplements in the admission interview. Therefore, we know that patients are using supplements randomly, however, not in a comprehensible and computable way. Therefore, dietary nutrient intake may also be higher due to supplement use among the patients. According to the biochemical parameters, we are aware, that the Friedewald formula (FF) used by our lab has certain limitations. Clinically, the most noteworthy limitation is that FF cannot be applied to samples with TG levels above 400 mg/d. Additionally, FF cannot be used in patients with dysbetalipoproteinemia (type III hyperlipoproteinemia), and when the patient is not fasting and therefore, chylomicrons are present. All these limitations were taken into consideration. Since we could exclude such cases, we used FF. Additionally, we are aware, that BIA (bioelectrical impedance analysis) or other defined measurements of body composition would have been preferable to BMI and WHR, since the latter are not adequately sensitive to determine nutritional status. However, the focus of the current study was to collect baseline information using a pragmatic approach to illuminate health behavior. Our study did not aim to reveal any associations between nutrition and disease-specific outcomes. However, this work followed a pragmatic approach to provide insight into nutritional and food choices with the goal of helping patients achieving a healthy diet.

## 5. Conclusions

This study provides an overview of the everyday eating habits of RA patients. Overall, the results show that many patients with RA do not meet dietary recommendations set by the German Nutrition Society, similar to the data of the German National Nutrition Survey II. On the other hand, most of the patients reported familiarity with dietary recommendations, but found it difficult implementing the recommendations into their diets. Adherence to practical dietary recommendations and individualized patient education may prevent or decrease the progression of RA. Further studies are required to investigate the impact of a healthy dietary intervention on long-term prognosis and future RA exacerbations. Such longer-term studies may help to clarify the impact of a healthy dietary approach, which could reduce reliance on drug therapy and the related adverse effects. Finally, further research is needed to assess the efficacy of a healthy nutrition intervention on improving diet quality in RA patients and developing disease-specific recommendations.

## Figures and Tables

**Figure 1 nutrients-15-00095-f001:**
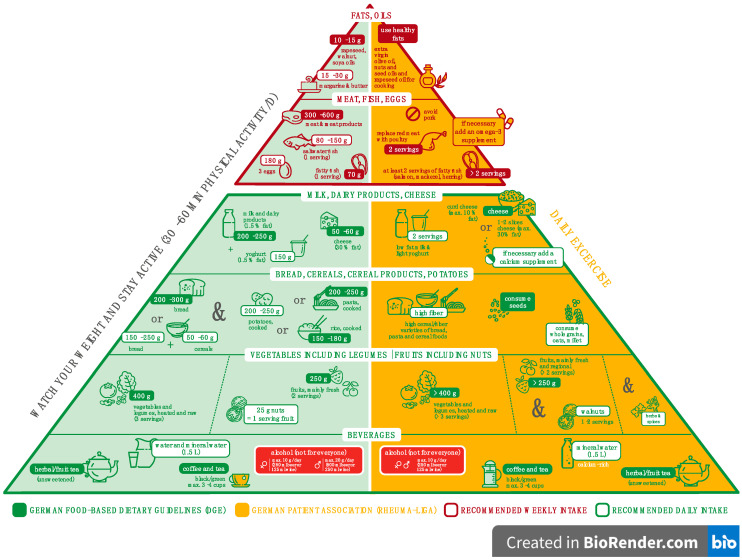
Food pyramid including German food-based dietary guidelines (DGE) and German patient association (Rheuma-Liga) recommendations for a wholesome diet.

**Table 1 nutrients-15-00095-t001:** Demographic data and current treatment of RA patients.

Variable	Unit	Female (*n* = 41)	Male (*n* = 20)
Demographic data			
n	%	67	33
Age *	years	65 (58–71)	57 (53.3–61.5)
		63.88 ± 10.81	58 ± 9.53
RA-specific data			
Disease duration *	years	3 (1.0–10)	0.8 (0.5–3.6)
		7.97 ± 10.95	4.94 ± 10.40
SDAI *	units	11.9 (6.1–20.1)	10.2 (7.1–15.1)
		13.64 ± 9.47	11.04 ± 5.71
CRP *	mg/dL	0.2 (0.1–0.4)	0.2 (0.1–0.5)
		0.5 ± 0.9	0.4 ± 0.5
Rheumatoid Factor IgM positive	*n* (%)	15 (37)	6 (30)
Anti-CCP-IgG antibody positive	*n* (%)	13 (32)	4 (20)
Anti-rheumatic treatment			
Methotrexate	*n* (%)	14 (34)	5 (25)
Other conventional (cs)-DMARDs	*n* (%)	6 (15)	1 (5)
Targeted synthesized (ts)-DMARDs	*n* (%)	2 (5)	1 (5)
Biologicals	*n* (%)	9 (22)	4 (20)
Glucocorticoids	*n* (%)	10 (24)	6 (30)

* Values are expressed as medians and interquartile range (IQR), means ± standard deviations (SD). SDAI, simple disease activity index; CRP, C reactive protein.

**Table 2 nutrients-15-00095-t002:** Anthropometric and biochemical parameters in patients with rheumatoid arthritis.

Variable	Unit	Female (*n* = 41)	Male (*n*= 20)
BMI *	kg/m^2^	25.7 (22.6–30.81)	27.8 (25.7–30.3)
		26.98 ± 5.62	28.12 ± 3.39
<18.5	*n* (%)	0	0
18.5–24.99	*n* (%)	19 (46)	4 (20)
25–29.99	*n* (%)	9 (22)	10 (50)
30–34.99	*n* (%)	8 (20)	5 (25)
35–39.99	*n* (%)	5 (12)	1 (5)
WHR *	W/H	0.9 (0.8–0.9)	1 (0.9–1)
		0.9 ± 0.1	1 ± 0.0
above normal values ^1^	*n* (%)	26 (63)	13 (65)
Waist circumference *	cm	94.5 (86.8–103.2)	105 (100–112.8)
		97.2 ± 14	106.4 ± 8.8
above normal values ^2^	*n* (%)	38 (93)	19 (95)
Biochemical data			
Total cholesterol *	mmol/L	6.08 (5.22–6.39)	5.34 (4.7–6.6)
		5.9 ± 1.1	5.5 ± 1.2
above normal values ^3^	*n* (%)	31 (76)	12 (60)
Fasting triglycerides *	mmol/L	1.28 (1.03–2.0)	1.37 (1.09–2.19)
		3.4 ± 1.6	3.9 ± 1.9
above normal values ^4^	*n* (%)	7 (17)	5 (25)
HDL-cholesterol *	mmol/L	1.76 (1.47–2.15)	1.33 (1.21–1.63)
		1.8 ± 0.4	1.5 ± 0.4
above normal values ^5^	*n* (%)	7 (17)	11 (55)
LDL-cholesterol *	mmol/L	3.36 (2.72–3.9)	3.09 (2.61–4.32)
		3.4 ± 1	3.3 ± 1.1
above normal values ^6^	*n* (%)	20 (49)	9 (45)
Non-HDL-cholesterol *	mmol/L	4.19 (3.41–4.73)	3.96 (3.25–5.0)
		4.1 ± 1.1	4.1 ± 1.2
above normal values ^7^	*n* (%)	31 (76)	13 (65)

* Values are expressed as medians and interquartile range (IQR), means ± standard deviations (SD). ^1^ ≥1.0 (men) and ≥0.85 (women); ^2^ ≥94 cm (men) and ≥80 cm (women); ^3,4^ <5.17 mmol/L; ^5^ >1.42 mmol/L; ^6^ <mmol/L; ^7^ >3.36 mmol/L. BMI, body mass index; WHR, waist to hip ratio; W/H; in this formula, W refers to the circumference of waist and H represents the circumference of the hip.

**Table 3 nutrients-15-00095-t003:** Energy and macronutrient intake in patients with RA (*n* = 61).

Macronutrient	Unit	Indexes	Female (*n* = 41)	Male (*n* = 20)	Recommendations *	*p*-Value
					Women/Men	
Energy	kcal/day	Median (IQR)	1557 (1193–1806)	1493 (1350–1932)	25–51 years: 1800/2300 ^1^	0.46
		Mean ± SD	1492 ± 417	1685 ± 588	51–65 years: 1700/2200 ^1^	
		% of norm	86.1	77.6	>65 years: 1700/2100 ^1^	
Carbohydrate	g/day	Median (IQR)	160 (128.1–189.8)	182.7 (150.8–205.3)	>50% Energy ^2^	0.12
		Mean ± SD	158.5 ± 49.6	189.5 ± 69		
		% of norm	73.2	68.9		
Of which sugar	g/day	Median (IQR)	27 (17.9–36.7)	28.3 (17.9–39)	max. 10% Energy ^3^	0.88
		Mean ± SD	30.8 ± 16.7	30.1 ± 17.9		
		% of norm	71.1	54.7		
Protein	g/day	Median (IQR)	54.5 (46.8–65.2)	63.9 (47.7–75.5)	25–51 years: 0.8 g/kg BW ^4^	0.07
		Mean ± SD	53.5 ± 16.3	63.7 ± 18.4	51–65 years: 0.8 g/kg BW ^4^	
		% of norm	114.5	116.2	>65 years: 1.0 g/kg BW ^4^	
Fat	g/day	Median (IQR)	63.6 (46.6–78.0)	56.5 (41.2–77.2)	30% Energy ^5^	0.64
		Mean ± SD	62.9 ± 21.9	63 ± 32.1		
		% of norm	109	86		
SFA	g/day	Median (IQR)	27.2 (18.4–34.3)	24.2 (17.1–32.1)	max. 10% Energy ^5^	0.47
		Mean ± SD	26.9 ± 9.9	26.1 ± 15.2		
		% of norm	140	107		
MUFA	g/day	Median (IQR)	20.7 (14–25.7)	18.1 (12.4–25.8)	7–10% Energy ^5^	0.49
		Mean ± SD	20.5 ± 8	19.3 ± 9.5		
		% of norm	125	93		
PUFA	g/day	Median (IQR)	8.3 (5.5–11)	8.3 (5.4–11.6)		0.86
		Mean ± SD	9.2 ± 5.5	9.8 ± 6.1		
LA	g/day	Median (IQR)	6.2 (4.38–8.48)	6.2 (4.5–9.1)	2.5% Energy ^5^	0.98
		Mean ± SD	7.3 ± 4.1	7.6 ± 5.5		
		% of norm	152	124		
ALA	g/day	Median (IQR)	0.8 (0.7–1.2)	0.8 (0.5–1)	0.5% Energy ^5^	0.45
		Mean ± SD	1.2 ± 1.3	0.9 ± 0.6		
		% of norm	125	74		
AA	mg/day	Median (IQR)	100 (70–140)	80 (48–120)		0.27
		Mean ± SD	127 ± 109	97 ± 76		
EPA	mg/day	Median (IQR)	0 (0–100)	0 (0–100)		0.84
		Mean ± SD	101 ± 199	159 ± 314		
DHA	mg/day	Median (IQR)	100 (0–300)	100 (0–200)		0.72
		Mean ± SD	171 ± 233	246 ± 426		

Values are expressed as medians and interquartile range (IQR), means ± standard deviations (SD), and as mean deviation from recommendations (% of norm). * D-A-CH reference values for nutrient intake issued by the Nutrition Societies of Germany, Austria and Switzerland. D-A-CH, arises from the initial letters of the common country identification for the countries Germany (D), Austria (A) and Switzerland (CH) and German Nutrition Society (DGE). SFA, saturated fatty acids; MUFA, monounsaturated fatty acids; PUFA, polyunsaturated fatty acids; LA, linoleic acid; ALA, linolenic acid; AA, arachidonic acid; EPA, eicosatetraenoic acid; DHA, docosahexaenoic acid. ^1^ D-A-CH 2015, based on physical activity level (PAL) 1.4; ^2,5^ D-A-CH 2000; ^3^ DGE 2018; ^4^ D-A-CH 2017.

**Table 4 nutrients-15-00095-t004:** Other nutrient intake in patients with RA (*n* = 61).

Others	Unit	Indexes	Female (*n* = 41)	Male (*n* = 20)	Recommendations	*p*-Value
					Women/Men	
Fiber	g/day	Median (IQR)	15.3 (11.2–19.6)	15.6 (10.8–21)	≥30 ^1^	0.8
		Mean ± SD	15.9 ± 7.1	16.7 ± 7.2		
		% of norm	53	55.5		
Insoluble fiber	g/day	Median (IQR)	10.8 (7.5–13.41)	9.8 (7.5–13.8)		0.88
		Min.–Max.	10.8 ± 5.1	11.1 ± 4.8		
Soluble fiber	g/day	Median (IQR)	4.9 (3.8–6.7)	5.2 (3.5–7.3)		0.65
		Mean ± SD	5.1 ± 2.2	5.5 ± 2.5		
Cholesterol	mg/day	Median (IQR)	232.7 (154.2–333.9)	205.4 (163.9–245.2)	300 ^2^	0.29
		Mean ± SD	249.7 ± 125.5	206.2 ± 92.5		
		% of norm	83.2	68.7		
Alcohol	g/day	Median (IQR)	0.1 (0–5.9)	1.2 (0–8.3)	10/20 ^3^	0.31
		Mean ± SD	4.4 ± 8.9	8.4 ± 14.6		
		% of norm	44	42		
Salt	g/day	Median (IQR)	5 (4–5.8)	4 (2.7–5.2)	6 ^4^	0.06
		Mean ± SD	4.1 ± 1.6	4.9 ± 1.4		
		% of norm	68.3	81.7		
Caffeine	mg/day	Median (IQR)	300 (124–600)	460 (335–802)	400 ^5^	0.04 *
		Mean ± SD	397 ± 337.9	638.7 ± 493.6		
		% of norm	99.3	159.7		

Values are expressed as medians and interquartile range (IQR), means ± standard deviations (SD), and as mean deviation from recommendations (% of norm). * D-A-CH reference values for nutrient intake issued by the Nutrition Societies of Germany, Austria and Switzerland. D-A-CH, arises from the initial letters of the common country identification for the countries Germany (D), Austria (A) and Switzerland (CH). ^1^ D-A-CH 2021; ^2^ German food-based dietary guidelines (DGE) 2010; ^3^ D-A-CH 2000; ^4^ D-A-CH 2020; ^5^ European Food Safety Authority (EFSA) 2015.

**Table 5 nutrients-15-00095-t005:** Selected micronutrient intake in patients with RA (*n* = 61).

Micronutrient	Unit	Indexes	Female (*n* = 41)	Male (*n* = 20)	Recommendations *	*p*-Value
					Women/Men	
Vitamin A	µg RE/day	Median (IQR)	900 (600–1300)	600 (400–900)	25–51 years: 700/850 ^1^	0.07
		Mean ± SD	1091.2 ± 834.6	786.7 ± 554.5	51–65 years: 700/850 ^1^	
		% of norm	155.9	94.4	>65 years: 700/800 ^1^	
Vitamin D	µg/day	Median (IQR)	1.6 (1.1–2.6)	1.5 (0.8–2.6)	20 ^2^	0.58
		Mean ± SD	2.8 ± 4.8	3.5 ± 5.5		
		% of norm	14	17.5		
Vitamin E	mg α-TE/day	Median (IQR)	8 (5.6–11.9)	6.8 (5.2–9.0)	25–51 years: 12/14 ^3^	0.27
		Mean ± SD	9 ± 4.6	8.2 ± 5.9	51–65 years: 12/13 ^3^	
		% of norm	76.9	63.1	>65 years: 11/12 ^3^	
Vitamin K	µg/day	Median (IQR)	55.5 (35.9–100.9)	43.3 (30.7–52.3)	25–51 years: 60/70 ^4^ >51 years: 65/80 ^4^	0.06
		Mean ± SD	113.6 ± 291.8	43.6 ± 16.2		
		% of norm	181.8	58.1		
Vitamin B6	mg/day	Median (IQR)	1 (0.8–1.2)	1.2 (0.8–1.4)	1.4/1.6 ^5^	0.2
		Mean ± SD	1 ± 0.4	1.2 ± 0.4		
		% of norm	71.4	75		
Folate	µg/day	Median (IQR)	171.9 (122.8–205.8)	140.1 (111.4–202)	300 ^6^	0.54
		Mean ± SD	169.3 ± 68.8	159.4 ± 63.3		
		% of norm	56.4	53.1		
Sodium	mg/day	Median (IQR)	1769.1 (1207.3–2214.9)	2214.6 (1733.1–2585.7)	1500 ^7^	0.04 *
		Mean ± SD	1803.8 ± 698.6	2195.3 ± 675.8		
		% of norm	120.3	146.4		
Calcium	mg/day	Median (IQR)	625 (456–779)	524 (419–736)	1000 ^8^	0.35
		Mean ± SD	625.1 ± 234.9	589.7 ± 303. 7		
		% of norm	62.5	59		
Magnesium	mg/day	Median (IQR)	241.6 (179.1–303.3)	269.8 (227.5–285.1)	300/350 ^9^	0.22
		Mean ± SD	243.4 ± 81.4	268.7 ± 66.1		
		% of norm	81.1	76.8		
Iron	mg/day	Median (IQR)	8.5 (7–11.4)	9.2 (7.9–11.1)	25–51 years: 15/10 ^10^ >51 years: 10 ^10^	0.58
		Mean ± SD	9 ± 3.2	9.6 ± 3		
		% of norm	72	96		
Zinc	mg/day	Median (IQR)	7 (5.4–9.3)	7.4 (4.6–9)	8/14 ^11^	0.9
		Mean ± SD	7.2 ± 2.5	7.3 ± 3		
		% of norm	90	52.1		
Copper	mg/day	Median (IQR)	1.4 (1.2–1.7)	1.6 (1.3–1.9)	1.0–1.5 ^12^	0.22
		Mean ± SD	1.5 ± 0.5	1.6 ± 0.5		
		% of norm	120	128		

Values are expressed as medians and interquartile range (IQR), means ± standard deviations (SD), and as mean deviation from recommendations (% of norm). * D-A-CH reference values for nutrient intake issued by the Nutrition Societies of Germany, Austria and Switzerland. D-A-CH, arises from the initial letters of the common country identification for the countries Germany (D), Austria (A) and Switzerland (CH). ^1^ D-A-CH 2020, retinol equivalent (RE); ^2^ D-A-CH 2012; ^3,4,10,12^ D-A-CH 2020, alpha-tocopherol equivalent (α-TE); ^5^ D-A-CH 2019; ^6^ D-A-CH 2018; ^7^ D-A-CH 2016; ^8^ D-A-CH 2013; ^9^ D-A-CH 2021;^11^ D-A-CH 2019, based on moderate phytate intake.

**Table 6 nutrients-15-00095-t006:** Consumption of food groups in patients with RA (*n* = 61).

Food Group	Unit	Indexes	Female (*n* = 41)	Male (*n* = 20)	Guidelines *	*p*-Value
Bread	g/day	Median (IQR)	106.7 (75.7–146.7)	174.8 (83.3–203.4)	150–300	0.02 *
		Mean ± SD	111.1 ± 50.7	162.6 ± 85.9		
		% of norm	49.4	72.3		
Cereals	g/day	Median (IQR)	23.3 (0–33.3)	33.3 (0–45.8)	50–60	0.5
		Mean ± SD	31.9 ± 43.1	31.7 ± 29.7		
		% of norm	58	57.7		
Potatoes, rice, pasta, total	g/day	Median (IQR)	50 (26.7–93.3)	33.3 (0–95.0)	150–250	0.61
		Mean ± SD	64.05 ± 53.9	66.3 ± 81.6		
		% of norm	32	33.2		
Vegetables, mushrooms and pulses	g/day	Median (IQR)	127.4 (60–225.3)	109.2 (66.7–185.8)	400	0.84
		Mean ± SD	151.86 ± 119.3	132 ± 91.3		
		% of norm	38	33		
Fruits and fruit products (without juice)	g/day	Median (IQR)	130 (86.7–236.7)	163.4 (65–203.2)	250	0.98
		Mean ± SD	210.6 ± 278.7	192.7 ± 241.5		
		% of norm	84.2	77.1		
Nuts (g/day)	g/day	Median (IQR)	0 (0–6.7)	0 (0–0)		0.12
		Mean ± SD	6 ± 10.4	1.8 ± 4		
Milk and dairy products	g/day	Median (IQR)	56.7 (11.7–101.7)	53.4 (0–80)	200–250	0.82
		Mean ± SD	71.3 ± 70.3	81.5 ± 103.2		
		% of norm	31.7	36.2		
Cheese	g/day	Median (IQR)	26.7 (13.3–40)	8.4 (0–47.1)	50–60	0.22
		Min.–Max.	30.8 ± 24.1	26.6 ± 33.4		
		% of norm	56	48.4		
Meat, meat products, sausages	g/week	Median (IQR)	466.9 (233.1–700)	577.5 (367.3–743.8)	300–600	0.17
		Mean ± SD	471.3 ± 318.2	578.4 ± 283.8		
		% of norm	104.7	128.5		
Fish, fish products, seafood	g/week	Median (IQR)	0 (0–163.1)	0 (0–326.7)	150–220	0.58
		Mean ± SD	97.2 ± 168.5	225.8 ± 367.4		
		% of norm	52.6	122.1		
Eggs	g/week	Median (IQR)	46.9 (0–140)	0 (0–140)	180	0.3
		Mean ± SD	114.1 ± 144.9	84 ± 131.7		
		% of norm	63.4	46.7		
Butter, margarine and oils	g/day	Median (IQR)	6.7 (0.7–11.7)	3 (0–6.2)	25–45	0.01 *
		Mean ± SD	11.2 ± 14.1	6.3 ± 9.6		
		% of norm	32	18		
Non-alcoholic beverages, total	ml/day	Median (IQR)	1805 (1333.4–2123.3)	1795.8 (1518.8–2283.3)	1500	0.83
		Mean ± SD	1768.6 ± 573.1	1820.9 ± 613.5		
		% of norm	117.9	121.4		
Confectionary, total	g/day	Median (IQR)	10 (5–36.7)	15 (5.8–37.7)		0.64
		Mean ± SD	23.3 ± 29.3	26.3 ± 30.5		
Pastries, total	g/day	Median (IQR)	66.7 (0–100)	62.9 ± 16.4		0.63
		Mean ± SD	64.8 ± 62.5	62.9 ± 73.3		

Values are expressed as medians and interquartile range (IQR), means ± standard deviations (SD), and as mean deviation from recommendations (% of norm). * German food-based dietary guidelines (DGE) 2017.

## Data Availability

Not applicable.

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
