# Peer review of "Nutrition Patterns and Their Gender Differences among Rheumatoid Arthritis Patients: A Descriptive Study"

_nutrients, 2022, doi:10.3390/nu15010095_

Round 1
Reviewer 1 Report
The paper is overall well-written and presented. To my opinion the two groups are not equivalent (male and female) so as to be compared as the RA severity and disease time are not comparable.
Nevertheless, it is of a common acceptance that the majority of patients (regardless to the disease) do not generally follow the strict dietary instructions of their physician. What would be interesting is to see the impact in long-term "wrong" diet to the RA symptoms or to the possibility of limiting the drug dose (e.g. methotrexate) due to adoption of a healthier diet. These observations should be included in the conclusions and suggest a broader study in the near future including also disease scores e.tc.
Author Response
Dear Reviewer,
thank you for your report and thoughtful comments on our manuscript.
Please see below our point-by-point response and the revised article attached.
Thank you for your second review in advance!
With best wishes,
Christina
Point 1: The paper is overall well-written and presented. To my opinion the two groups are not equivalent (male and female) so as to be compared as the RA severity and disease time are not comparable.
Response 1: Thank you very much for this kind comment, indeed men and women are not directly comparable; however, the main purpose of the study was to look at dietary habits of people with RA and we thought it was important to report the differences in dietary habits between men and women.
Point 2: Nevertheless, it is of a common acceptance that the majority of patients (regardless to the disease) do not generally follow the strict dietary instructions of their physician. What would be interesting is to see the impact in long-term "wrong" diet to the RA symptoms or to the possibility of limiting the drug dose (e.g. methotrexate) due to adoption of a healthier diet. These observations should be included in the conclusions and suggest a broader study in the near future including also disease scores e.tc.
Response 2: Thank you for this important comment as well. A study evaluating the effects of a healthy diet intervention versus an unhealthy diet is currently running at our center. The patients in our manuscript were enrolled in a trial spanning a total of 16 weeks of intervention with supplements that would improve their typically unhealthy diet to a diet with a healthier composition and then re--analysis of the data from this ongoing study will take place. So, from this group we cannot evaluate the long-term effects of “wrong” dieting, but the impact of the trial scheme on reduction of drug doses will be determined in the final analysis. Therefore, in response to your comments, we have added to our conclusions highlighted in red, see line 338, revised article attached).
Conclusions
This study provides an overview of the everyday eating habits of RA patients. Overall, the results show that many patients with RA do not meet dietary recommendations set by the German Nutrition Society, similar to the data of the German National Nutrition Survey II. On the other hand, most of the patients reported familiarity with dietary recommendations, but found it difficult implementing the recommendations into their diets. Adherence to practical dietary recommendations and individualized patient education may prevent or decrease the progression of RA. Further studies are required to investigate the impact of a healthy dietary intervention on long-term prognosis and future RA exacerbations. Such longer-term studies may help to clarify the impact of a healthy dietary approach, which could reduce reliance on drug therapy and the related adverse effects. Finally, further research is needed to assess the efficacy of a healthy nutrition intervention on improving diet quality in RA patients and developing disease-specific recommendations.

Reviewer 2 Report
Relocate the figures to places where they are mentioned/discussed in the text. In general, the tables seem to be randomly placed in the text… table 1 - demographic data should not be in nutritional education paragraph; table 2 – anthropometric and biochemical parameters placed in statistical analysis paragraph?! Also, they lack proper explanation (abbreviations e.g. DACH, EFSA…). Figure 1 is complex, but difficult to read. In addition, place the Tables in order.
Line 75 – how was it double-blind?
BMI and WHR are not sufficient to characterize nutritional status of the patients.
Line 104 – what are PAP-methods?
Usage of Friedewald formula has its limitations (doi: 10.7860/JCDR/2015/16850.6870). Were the levels of triglycerides taken into consideration, when using this formula?
How did you test normal distribution of the data? Using t-test?
Why some data are presented as means with SD and some with SEM? E.g. in the text (line 154), there is value with what? SEM? SD?
Tables 3 and 4. Does it make a sense to statistically compare male vs female intake of nutrients (in absolute values) when they have different standards? Was the nutrient intake normalized to sex? Or just % of norm was compared?
All tables are missing legends.
Were all of the data normally distributed (Arachidonic acid, Eicosapentaenoic acid and Docosahexaenoic acid look non-Gaussian distributed)? If not, median + IQR should be presented.
Line 225 - There is a statement: “For this study, 225 we did not include supplements in the food records.“ Can you discuss why?
In case of available selected anthropometric and biochemical parameters, the authors could them corelate with nutritional status - despite they mentioned that it was not the aim. The finding that RA patients are typical of same mistakes in healthy nutrition as an average western population is not sufficient finding to warrant the publication.
Author Response
Dear Reviewer,
thank you for your report and thoughtful comments on our manuscript.
Please see below our point-by-point response and the revised article attached.
Thank you for your second review!
With best wishes,
Christina
Point 1: Relocate the figures to places where they are mentioned/discussed in the text. In general, the tables seem to be randomly placed in the text… table 1 - demographic data should not be in nutritional education paragraph; table 2 – anthropometric and biochemical parameters placed in statistical analysis paragraph?! Also, they lack proper explanation (abbreviations e.g. DACH, EFSA…). Figure 1 is complex, but difficult to read. In addition, place the Tables in order.
Response 1: Thank you very much for this constructive comment. We have updated the numerical arrangement of the figure and all tables. We have included abbreviations for the tables. In addition, we have maximized figure 1 to make it reader friendly. Please see revised article attached (in red).
Table 1. Demographic data and current treatment of RA patients; new line 117.
Table 2. Anthropometric and biochemical parameters in patients with rheumatoid arthritis; new line 142.
Table 3. Energy and macronutrient intake in patients with RA (n = 61); new line 193.
Table 4. Selected micronutrient intake in patients with RA (n = 61); new line 206.
Table 5. Other nutrient intake in patients with RA (n = 61); new line 212.
Table 6. Consumption of food groups in patients with RA (n = 61); new line 224.
Figure 1. Food pyramid including German food-based dietary guidelines (DGE) and German patient association (Rheuma-Liga) recommendations for a wholesome diet; new line 52.
Point 2: Line 75 – how was it double-blind?
Response 2: Indeed, the paragraph may be misleading. The patients described here were from a study group in an ongoing trial with a nutritional supplement intervention, which was in a placebo-controlled double-blind design. To avoid misunderstandings, we have changed the expression to:
“The patients participating in this analysis were screened at enrollment in an ongoing single-centre randomized, double-blind, controlled clinical study that conformed to the Declaration of Helsinki....”, please see new line 108.
Point 3: BMI and WHR are not sufficient to characterize nutritional status of the patients.
Response 3: We are aware that BMI and WHR are not the gold standard for characterization for the nutritional status; however, it is common practice to use these factors at the trial location. Since this study aimed to collect baseline information on nutritional habits and provision of nutritional education, rather than to reveal any associations between nutrition and outcomes, we have investigated only the BMI and WHR. But your comment is justified, therefore, we have added a comment on this issue in the discussion section, highlighted in red; see new line 326.
Also, we are aware, that BIA (bioelectrical impedance analysis) or other defined measurements of body composition would have been preferable to BMI and WHR, since the latter are not adequately sensitive to determine nutritional status. However, the focus of the current study was to collect baseline information using a pragmatic approach to illuminate health behavior.
Point 4: Line 104 – what are PAP-methods?
Response 4: These are common routine lab methods for the indirect quantitation of Cholesterol and Triglycerides via Phenol and Aminophenazone (PAP). We have added this full test name; see line 138.
Point 5: Usage of Friedewald formula has its limitations (doi: 10.7860/JCDR/2015/16850.6870). Were the levels of triglycerides taken into consideration, when using this formula?
Response 5. We are aware that Friedewald formula (FF) has certain limitations. Clinically, the most noteworthy limitation is that FF cannot be applied to samples with TG levels above 400 mg/d. Also, FF cannot be used in patients with dysbetalipoproteinemia (type III hyperlipoproteinemia) and when chylomicrons are present. All these limitations were taken into the considerations. Since we could exclude such cases, we used FF, however we have added this aspect in the discussion section; see line 322.
Point 6: How did you test normal distribution of the data? Using t-test?
Response 6: Thank you for this suggestion. Indeed, we did not test the data for normal distribution using t-test. Normality of the analyzed variables were tested with the Kolmogorov-Smirnov test. We have corrected this information, see line 169 - 171.
Point 7: Why some data are presented as means with SD and some with SEM? E.g. in the text (line 154), there is value with what? SEM? SD?
Response 7: Due to non-normality some data were presented as Median +/- IQR. In response to the reviewer’s suggestion, we have converted all data statistics into Median values +/- Quartile and means ± standard deviation. Values in line 154 (3.2 Dietary Intake) now presented as mean ± standard deviation; new line 182.
Point 8: Tables 3 and 4. Does it make a sense to statistically compare male vs female intake of nutrients (in absolute values) when they have different standards? Was the nutrient intake normalized to sex? Or just % of norm was compared?
Response 8: The comparison is based on recommended intake by gender and consequently on deviation from recommendations. Tables were updated for better visualization.
Point 9: All tables are missing legends.
Response 9: We have added legends to all tables.
Point 10: Were all of the data normally distributed (Arachidonic acid, Eicosapentaenoic acid and Docosahexaenoic acid look non-Gaussian distributed)? If not, median + IQR should be presented.
Response 10: Please see response No 7. Data are all presented also as median + IQR.
Point 11: Line 225 - There is a statement: “For this study, 225 we did not include supplements in the food records.“ Can you discuss why?
Response 11: None of the patients recorded in the 3-day-food records any intake of supplements. But we asked yes-no-questions about the intake of vitamins and mineral supplements. Therefore, we know that patients are using supplements. Detailed information on which kind of supplements and dosage was not collected at that time.
We have included these information in the discussion section, highlighted in red; see line 318:
We could not rate the consumption of supplements in this study, since none of the patients re-corded any intake of supplements in the 3-day-food records. In addition, we asked yes-no questions about the intake of vitamins and mineral supplements in the admission interview. Therefore, we know that patients are using supplements randomly, however, not in a comprehensible and computable way. Therefore, dietary nutrient intake may also be higher due to supplement use among the patients.
Point 12: In case of available selected anthropometric and biochemical parameters, the authors could them corelate with nutritional status - despite they mentioned that it was not the aim. The finding that RA patients are typical of same mistakes in healthy nutrition as an average western population is not sufficient finding to warrant the publication.
Response 12: This was a descriptive study to collect baseline information on nutritional habits as well as the occurrence of nutritional education. This study aimed to provide an overview of the everyday eating habits of RA patients receiving standard treatment. Present findings show a poor handling of patient education regarding a healthy diet in the context of secondary prevention, which should be an important component of adequate therapy. A study evaluating nutrition and outcomes is currently running at our center, which will provide more information that will allow us to evaluate the effect of the healthier diet intervention on nutritional status as well as other RA outcomes. Of course, everyone could benefit from education on a healthier diet. However, the point of this report is that, even though their dietary habits reflect that of the general population, people with RA have a chronic condition that could respond in a positive way to a healthy diet, but they are not receiving individualized specific nutrition education as part of their treatment. This is a missed opportunity on the part of rheumatologists and the support staff caring for people with RA. The ongoing study could help to determine whether improved nutrition education affects outcomes in people with RA.
If you agree, we would like to update the title of this article to:
Nutrition patterns and their gender differences among rheumatoid arthritis patients: a descriptive study.

Reviewer 3 Report
The authors well explored the nutrition patterns and their gender differences among rheumtoid arthritis patients. I have one suggestion: The specific information about the nutritional education should be included in the part 2.6.
Author Response
Dear Reviewer,
thank you for your report and thoughtful comments on our manuscript.
Please see below our point-by-point response and the revised article attached.
Thank you for your second review!
With best wishes,
Christina
Point 1: The authors well explored the nutrition patterns and their gender differences among rheumatoid arthritis patients. I have one suggestion: The specific information about the nutritional education should be included in the part 2.6.
Response 1: Thank you very much for this kind comment. We have included the specific information about the nutritional education, highlighted in red (see line 163, revised article attached):
2.6. Nutritional Education
Aside from dietary intake, the patients were asked whether they had already received information about a healthy diet as part of their disease treatment plan, and, if so in what context. Possible answers were “yes” or “no”, based on medical advice, detailed advice from a nutritionist, advice from Rheuma-Liga by written information e.g. brochures, leaflets. Specific information on nutritional education, based on guidelines provided by the Rheuma-Liga, describes a healthy diet to include vegetables, fruits, whole grains, nuts, seeds, herbs, spices, omega-3-rich foods (e.g. fish and healthy oils), and a diet low in saturated fat, arachidonic acid from red meat and milk products, and refined carbohydrates, as shown in Figure 1.
In addition, we have maximized figure 1 to make it reader friendly; please see line 52, revised article attached.

Round 2
Reviewer 1 Report
The points have been addressed and the document adequately revised. No other comment from my side.
Author Response
Dear Reviewer,
Thank you for your comments and feedback to us.
We hope our findings will deserve further trials to increase awareness and patient education to help patients achieve optimal dietary intake.
Once again thank you!
With best wishes,
Christina
Reviewer 2 Report
In the revised version - Figure 1 is better readable. However, the consumption of wine or beer is allocated in the section „Non-alcoholic beverages“. Additionally, the units are too small: 0,25 ml of beer? Or 0,125 ml of wine???
I suggest the authors at least correlate dietary pattern with data presented in Table 1 and Table 2. The conclusion that „patients with RA do not meet dietary recommendations“ is not surprising, but well known.
Author Response
Dear Reviewer,
again thank you for your constructive comments and suggestions.
Please see below our response.
Thank you for your review in advance.
With best wishes,
Christina
Point 1: In the revised version - Figure 1 is better readable. However, the consumption of wine or beer is allocated in the section „Non-alcoholic beverages“. Additionally, the units are too small: 0,25 ml of beer? Or 0,125 ml of wine???
Response 1: Thank you very much for this kind comment, indeed the section with regards to beverages is not clear and too small. We have updated and maximized the section with regards to beverages, please see attached.
Point 2: I suggest the authors at least correlate dietary pattern with data presented in Table 1 and Table 2. The conclusion that „patients with RA do not meet dietary recommendations“ is not surprising, but well known.
Response 2: To the best of our knowledge, no previous study has described nutritional habits and occurrence of nutritional education in patients with RA, living in Germany. Therefore, it was important for us to collect these data. To investigate the relationship/association of anthropometric and biochemical parameters with nutritional patterns was not the aim of the study. Our results indicate that RA patients have better nutritional status than the general population in Germany, because they are familiar with nutrition recommendations provided by their rheumatologist. The key problem that we found in our descriptive study is that there is a need for improvement in nutrition education since most patients were not sufficiently informed. This can be seen as missed opportunity. It appears reasonable to use every single chance to increase awareness and patient education to help patients achieve optimal dietary intake.
We already had performed the correlations (Spearman’ s correlation test was used for correlations) of the dietary patterns with the data presented in table 1 and 2, however, due to study objective and few (well known) significant findings, we did not include the findings in our original manuscript.
Due to your suggestion, we now have added the data of the correlation to Supplementary Materials (see line 307), to statistical analysis (see new line 134) and results (see line 193).
